

# Mammals in the Chornobyl Exclusion Zone's Red Forest: a motion activated camera trap study

Nicholas A. Beresford[1,3], Sergii Gashchak[2], Michael D. Wood[3], Catherine L. Barnett[1]

[1]UK Centre for Ecology & Hydrology, Lancaster Environment Centre, Bailrigg, Lancaster, LA11 4AP, UK

[2]Chornobyl Center for Nuclear Safety, Radioactive Waste & Radioecology, International Radioecology Laboratory, 77[th] Gvardiiska Dyviiya Str.11, P.O. Box 151, 07100 Slavutych, Kyiv Region, Ukraine

[3]School of Science, Engineering & Environment, University of Salford, Manchester, M5 4WT, UK

*Correspondence to*: Nicholas A. Beresford (nab@ceh.ac.uk)

**Abstract**

Since the accident at the Chornobyl nuclear power plant in 1986 there have been few studies published on medium/large mammals inhabiting the area from which the human population was removed (now referred to as the Chornobyl Exclusion Zone). The dataset presented in this paper describes a motion activated camera trap study (n=21 cameras) conducted from September 2016 - September 2017 in the Red Forest located within the Chornobyl Exclusion Zone. The Red Forest, which is likely the most anthropogenically contaminated radioactive terrestrial ecosystem on earth, suffered a severe wildfire in July

2016. The motion activated trap cameras were therefore in place as the Red Forest recovered from the wildfire. A total of 45859 images were captured and of these 19391 contained identifiable species or organism types (e.g. insects). A total of 14 mammal species were positively identified together with 23 species of birds (though birds were not a focus of the study).

Weighted absorbed radiation dose rate rates were estimated for mammals across the different camera trap locations; the

number of species observed did not vary with estimated dose rate. We also observed no relationship between estimated weighted absorbed radiation dose rates and the number of triggering events for the four main species observed during the study (Brown hare, Eurasian elk, Red deer, Roe deer).

The data presented will be of value to those studying wildlife within the CEZ both from the perspectives of the potential

effects of radiation on wildlife and also rewilding in this large, abandoned area. They may also have value in any future studies investigating the impacts of the recent Russian military action in the CEZ.

The data and supporting documentation are freely available from the Environmental Information Data Centre (EIDC) under the terms and conditions of a Creative Commons Attribution (CC BY) licence: https://doi.org/10.5285/bf82cec2-5f8a-407c-

bf74-f8689ca35e83 (Barnett et al. 2022a).



## 1 Introduction

Following the 1986 Chornobyl nuclear power plant accident coniferous trees up to 4 km to the west of the reactor were killed by radiation over an area of approximately 4-6 km$^2$ (coniferous trees covered approximately 40 % of this area in 1986 (Kyiv Politech Institute's Museum 2022)). The area is now known as the 'Red Forest' and it is likely the most

anthropogenically contaminated radioactive terrestrial ecosystem on earth. It has subsequently regenerated with understorey vegetation and, to some extent, with deciduous trees. Whilst over the years many studies have been conducted within the Red Forest (e.g. Geras'kin et al. 2008; Møller & Mousseau 2013; Møller et al. 2016;  Lavrinienko et al. 2018 a,b; Antwis et al. 2021; Beresford et al. 2022) none has studied utilisation of the area by medium/large mammals. From the study site map within the original paper of Møller & Mousseau (2013) their mammal snow track study did include sites in the Red Forest,

as well as other sites in the CEZ; this study concluded that the abundance of mammals decreased with increasing radiation.

In July 2016 there was a severe fire within the Red Forest with approximately 80% of the forest being burnt (Beresford et al. 2021). In September 2016 as one of a number of studies (Antwis et al. 2021; Beresford et al. 2021; Jackson et al. in preparation) considering the effects of and recovery from the fire, we set-up a network of 21 motion activated camera traps

across the Red Forest which were left in place to record primarily medium/large mammals for a period of approximately one year. This paper describes and discusses this study; all the photographs are freely available from https://catalogue.ceh.ac.uk/documents/bf82cec2-5f8a-407c-bf74-f8689ca35e83 (Barnett et al. 2022a).

## 2 Materials and Methods

### 2.1 Motion activated digital trap camera deployment

Twenty-one Little Acorn 6210MC motion activated digital trap cameras, fitted with 8 Gb memory card to record images, were installed across the Red Forest in early September 2016; the cameras were operated for approximately a year until September 2017. The cameras were deployed using an approximate grid pattern with three rows of seven cameras (see Fig. 1). No bait was used to attract animals.

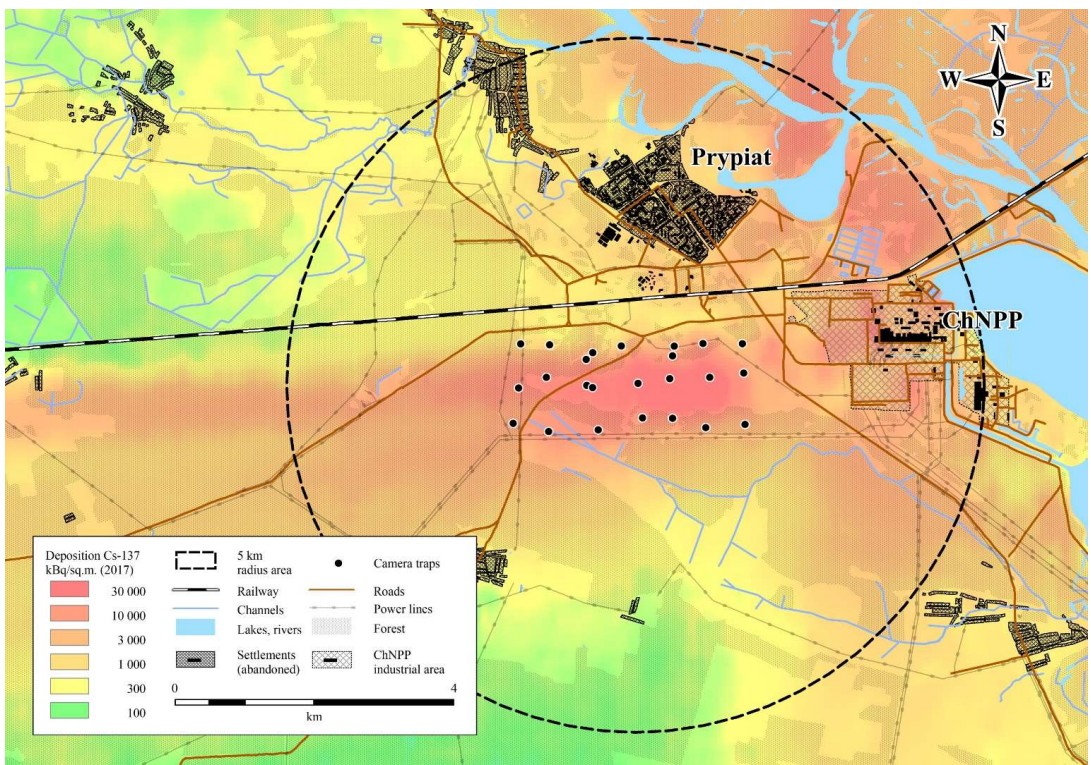

**Figure 1: Map showing the location of the study cameras overlaid on a $^{137}$Cs deposition surface (decay corrected to 2017). The large circle is the 5 km radius area over which absorbed weighted dose rates were calculated. Figure produced by and published with the permission of the Chornobyl Center.**

When deploying each camera for the first time, approximately 20 poles (1 m high with markings at every 20 cm) were positioned in front of the camera in three parallel rows one metre apart; each row began three metres in front of the camera and ended eight metres away from the camera. The camera was then activated to capture an image of the poles *in situ,* and the poles were then removed (an image of the pole positions at sites where these were recorded has been included within the dataset associated with this study (Barnett et al., 2022a); some of these images contain images of co-authors with their

permission). The images of the poles can be used to estimate animal height and distance from the camera should this be desired. Tree branches, tall grasses and bushes that were likely to obscure the camera or cause false activation by their movement were cleared from an area of about 40-60 m$^2$ in front of the camera at the initial set-up and when necessary, throughout the study.



Each camera was mounted at a height of approximately 0.7 m (typically attached to trees) to principally record images of medium/large mammals although images of small mammals, birds and occasional insects were also captured. The cameras were positioned such that they mostly faced north to shelter them from false activation caused by direct sunlight. When triggered by movement all the cameras were pre-set to take a three-image burst; the interval between these three images was <1 second. The time delay between one three-burst cycle and any immediate subsequent cycle was approximately 2-4

seconds; it was therefore possible that some animals may not have been captured if they were moving rapidly across the field of view during this time. All the cameras were capable of capturing images both day and night (and during the transition period in between) by using an infrared sensor and invisible infrared flash (850 or 940 nm, capable of lighting an area of up to 10 m in front of the camera); the appropriate day/night/transition setting is subjective as the camera automatically choses the appropriate day/night/transition setting based on light level. All the cameras were inspected, and the data (images and

image metadata) downloaded from the memory cards on three occasions during the study (March 2017, June 2017 and September 2017), these are referred to as setup 1-3 in the accompanying dataset; the cameras were also randomly inspected throughout the year to check functionality (and to ensure they had not been stolen). The images and image metadata were supplied to UKCEH by Chornobyl Center as .jpeg. .avi and MSExcel files, respectively. The image catalogue described in Sect. 2.3 was then populated by UKCEH using these files.


Information related to each camera and each deployment period has been provided in file 'REDFIRE_Trap_Camera_Details_And_Image_Summary' which is included within the dataset associated with this study (Barnett et al., 2022a). The information provided includes: location (site number); numerical camera identifier; setup number (1, 2 or 3, see above); start date and time and end date; and time of each deployment period (most cameras were set to record

at Eastern European summer time throughout with the exception of cameras 155 and 156 during setup 2 which were set to record at Eastern European winter time in error and cameras 161 and 174 from setup 1 where the time shown on the image was recorded incorrectly, the data related to date and time has therefore been manually corrected within the image catalogue to Eastern European summer time for these four cameras); the total number of days each camera was in-use during each deployment period; and any notes relevant to the cameras or their operation.


At sites 163, 168 and 169 the trap cameras were stolen during setup 1 and therefore no images from these cameras were recovered for that setup. The cameras were replaced at the start of setup 2 with new cameras located at nearby sites 362, 364 and 365, respectively. During setup 2 cameras from sites 157, 164, 170, 175 and 362 were stolen and not replaced and the memory cards from cameras 158 and 171 were changed part way through; during setup 3 camera 158 did not operate. In

total, images were recovered from 18 cameras for setup 1, 16 cameras for setup 2 and 15 for setup 3. The camera located at site 172 was set to record video in error during setup 1 at a service visit in late October (photographs were recorded



September to October as for the other cameras); the videos (20 seconds each) are included in the image catalogue and have been analysed in the same way as the photographs (see Sect. 2.3).

### 2.2 Study site and site characteristics

The site descriptive parameters, recorded by the same person for every site in early September 2016, include: numerical site identifier and location (latitude and longitude, WGS84); ambient dose rate measured at a height 1 m above soil surface; an evaluation of the fire damage as visible in September 2016 ('none', 'low and 'medium' and 'high')  together with an estimate of the percentage of the area within 100 m of the site affected by the fire; an estimation of the density of grassy vegetation and undergrowth over a 20 m radius of the camera location; notes on habitat within a 100 m radius of the camera

location; the dominant (>80%) tree species present and the approximate age of trees within a 100 m radius of the site and; the presence (or absence) of animal trails/tracks or water sources within 20 m of the site. The dataset also contains an estimate of the Cs-137 and Sr-90 soil activity concentrations (kBq m$^{-2}$) averaged over a 500 m radius centred on the camera site estimated from a spatial dataset (Shestopalov, 1996) and decay corrected to 1$^{st}$ March 2017. The Shestopalov (1996) data are presented as Bq m$^{-2}$; to convert to Bq kg$^{-1}$ we assumed a soil bulk density of 1.14 g cm$^{-3}$ dry mass estimated from data for

the Red Forest (Barnett et al. 2021) assuming a 10 cm soil depth as required for the subsequent estimation of estimated weighted absorbed dose rates (see below). This information is provided in the file 'REDFIRE_Trap_Camera_Site_Descriptions' within the dataset associated with this study (Barnett et al., 2022a).

### 2.3 Image catalogue

The image catalogue contains a description of information related to each image. The majority of the images obtained have

been included within the dataset associated with this study (Barnett et al., 2022a). However, to protect privacy any images containing people have not been included, although observations of people (other than members of the research team setting up and servicing the cameras) have been recorded in the catalogue. For cataloguing the images, a triggering event was assumed to begin when the camera motion sensor was triggered by an animal. A new triggering event was not assumed until at least 90 seconds had elapsed since an animal was last observed. However, there may be longer time periods between

triggering events where images are obviously part of the same sequence (e.g. an animal lays down for a period of time).

Within the dataset associated with this study (Barnett et al., 2022a) all the images (including those which did not capture any animal) are located within three sub-folders called REDFIRE_Setup_1, REDFIRE_Setup_2, REDFIRE_Setup_3 and within each of these folders are multiple sub-folders (with the format e.g. 'Setup1_Site155_2317') which correspond to the

'Image_Location_Folder_Name' column within the image catalogue described below. Within each of these sub-folders are further sub-folders entitled the common species names of animals observed. The individual images of each animal are located within these folders and are supplied as .jpg files and have the format e.g. IMAG0016. As noted in Sect. 2.2, at site





172 the camera was set, at a service visit in October, to record videos in error; the text 'Video' has been used within the notes column of the image catalogue to identify where video rather than photographs were recorded (camera 172 setup 1

only); the videos have been provided within the dataset.

In the file 'REDFIRE_Trap_Camera_Image_Catalogue', which is included within the dataset associated with this study (Barnett et al., 2022a), each image record (row) within the catalogue gives details of: setup number (1, 2 or 3, see Sect. 2.2), location (site number); numerical camera identifier; image location folder name (see above, e.g. Setup1_Site155_2317);

image filename (e.g. IMAG0127); date, time and period of the day (day, night, transition) related to when the image was captured; the common name of species captured in the image; the number of animals visible in the image; the number of animals seen per triggering event (cumulative; the triggering event number is recorded as 'n/a' for observations of people); triggering event number (sequential); the temperature when image was captured (°C, recorded by the camera at the start of each new triggering event (note this measurement is indicative only and not an absolute value (e.g. direct sun on the camera

affects the temperature recorded)); A marker ('Y') identifying the start of each new triggering event; A marker ('Y') identifying if an obviously young animal is present within the image (this is subjective and may not always have been noted) and; notes relating to the image (e.g. two species present within the image (where this occurs the data for the image is entered twice, once for the first species and again for the second species; the second species is allocated a new triggering event number). If the image was too poor to definitely identify the animal, the species common name has been recorded as

'Unidentifiable' occasionally for such images the potential species/animal type has been entered into the notes column. Images containing no images of animals are included within the dataset associated with this study; these are catalogued separately.

### 2.4 Quality control

Data were entered into the image catalogue by UKCEH staff (who were not aware of the comparative contamination levels

at the different camera sites), these data were then compared to a second set of data entered into a second catalogue by staff at the Chornobyl Centre for Nuclear Safety; any disparities were investigated and amended manually where necessary. Once this check was completed a final check was conducted by further UKCEH staff to ensure the information within the catalogue matched the images included within the dataset.

### 2.5 Estimation of total weighted absorbed dose rate

Indicative weighted absorbed dose rates have been estimated for example mammals in the study area using the ERICA Tool (v2.0; Brown et al., 2016). As inputs to the dose estimation, the $^{137}$Cs and $^{90}$Sr soil activity concentrations estimated for a 500 m radius around each camera site (see above) were used. This area equates to the potential home range of Brown hare which is likely the species with the smallest home range of the most commonly observed mammals (Schai-Braun & Hackläder



2014; CABI 2013). However, it is unlikely that the majority of mammal species observed would spend all of their time
within the relatively small area of the Red Forest. Therefore, [137]Cs and [90]Sr soil activity concentrations were estimated over
an area with a radius of 5 km, centred on the middle of the Red Forest, which may be appropriate for the larger species
observed (e.g. Okarma et al., 1998; Ofstad et al., 2016). All soil radionuclide activity concentrations were decay corrected to
01/03/2017 (approximately the midpoint of the study). The ERICA Tool contains a default terrestrial organism 'Mammal –
large' with dimensions equating to a large deer species (mass 245 kg) and dose rates were estimated for this default
organism. For comparison, an organism was created in the ERICA Tool equating to a Red fox, a regularly observed smaller
species which may spend part of its time underground (assumed dimensions for the Red fox were 0.4 x 0.15 x 0.2 m with a
mass of 6.6 kg (Pröhl. 2003)). The Large mammal geometry was assumed to spend 100 % of its time on the ground surface
and the Red fox 10 % of time underground (Brown, et al., 2003). The probabilistic Tier 3 of the ERICA Tool was used
inputting mean and standard deviation soil activity concentrations. The default mammal concentration ratios (and associated
probability distribution functions) in the ERICA Tool were used to estimate whole-body radionuclide activity concentrations
of the animals and consequently the internal dose rate; default radiation weighting factors of 3 for low energy beta emissions
and 1 for other beta and gamma emissions were used. The resultant mean, variance and median estimates of total weighted
absorbed dose rates were recorded for each of [137]Cs and [90]Sr (all estimates are presented in µGy h[-1] in Barnett et al., 2022a).

**3 Overview of images included within the catalogue**

A total of 45857 images were captured (not including photographs recorded during camera set-up and servicing), of these
19391 contained identifiable species or organism types (e.g. insects), 565 recorded people, 349 were of poor quality such
that the species could not be determined and 25552 images recorded no animals (i.e. predominantly false triggers due to
vegetation movement, light etc. or potentially a triggering by an animal that was not captured). A total of 14 mammal species
were positively identified together with 23 species of birds (Table 1).

Table 1. Species captured on the motion activated digital trap cameras.

| Common species name | [a]Latin species name |
| --- | --- |
| **Mammals** | |
| Brown hare | *Lepus europaeus* |
| Eurasian elk | *Alces alces* |
| Eurasian lynx | *Lynx lynx* |
| European badger | *Meles meles* |
| Domesticated dog (feral) | *Canis lupus familiaris* |
| Grey wolf | *Canis lupus* |



| Common species name | [a]Latin species name |
| --- | --- |
| Marten sp. | Martes (Genus) |
| [b]Mouse sp. | Muridae (Family) |
| Przewalski's horse | *Equus ferus przewalskii* |
| Raccoon dog | *Nyctereutes procyonoides* |
| Red deer | *Cervus elaphus* |
| Red fox | *Vulpes vulpes* |
| Red squirrel | *Sciurus vulgaris* |
| Roe deer | *Capreolus capreolus* |
| Wild boar | *Sus scrofa* |
| [c]Unidentifiable | Not applicable |
| **Birds** | |
| Black grouse | *Lyrurus tetrix* |
| Common blackbird | *Turdus merula* |
| Common buzzard | *Buteo buteo* |
| [b]Common quail | *Coturnix coturnix* |
| [b]Common snipe | *Gallinago gallinago* |
| Common wood pigeon | *Columba palumbus* |
| [b]Corncrake | *Crex crex* |
| Eurasian bittern | *Botaurus stellaris* |
| Eurasian hoopoe | *Upupa epops* |
| Eurasian Jay | *Garrulus glandarius* |
| Eurasian sparrowhawk | *Accipiter nisus* |
| Eurasian woodcock | *Scolopax rusticola* |
| European nightjar | *Caprimulgus europaeus* |
| European robin | *Erithacus rubecula* |
| Fieldfare | *Turdus pilaris* |
| Finch sp. | Fringillidae (Order) |
| Great egret | *Egretta alba* |
| Great grey shrike | *Lanius excubitor* |
| Great spotted woodpecker | *Dendrocopos major* |
| Great tit | *Parus major* |
| Hazel grouse | *Tetrastes bonasia* |



| Common species name | <sup>a</sup>Latin species name |
| --- | --- |
| <sup>b</sup>Marsh tit | *Poecile palustris* |
| Mistle thrush | *Turdus viscivorus* |
| Red backed shrike | *Lanius collurio* |
| Shrike sp. | *Lanius sp.* |
| Song thrush | *Turdus philomelos* |
| Thrush sp. | *Turdus sp.* |
| <sup>c</sup>Unidentifiable bird | Not applicable |
| **Other species** | |
| Unidentifiable insect | Insecta (Class) |
| Butterfly or Moth | Lepidoptera (Order) |
| Dragonfly | Odonata (Order) |
| Spider | Araneae (Order) |

<sup>a</sup>In some instances animals are identified at the Class, Order, Family level only. <sup>b</sup>Species is only mentioned within the notes column of the image catalogue (i.e. as a potential but not definitive observation). <sup>c</sup>Mammal or bird which could not be positively identified at species/genus level.


A summary of the images within the catalogue (e.g. number of images with mammals, birds or insects, number of images with nothing in, number of images with people in) and the total number of triggering events recorded (by setup, by site, by camera) has been provided in the file 'REDFIRE_Trap_Camera_Details_And_Image_Summary' within the dataset associated with this study (Barnett et al., 2022a). The dataset also provides a summary for mammals (filename: 'REDFIRE_Trap_Camera_Summary_Mammals'), by species, by camera and by setup of the number of triggering events and the mean, minimum and maximum of the number of individuals recorded per triggering event. For ease of comparing across setups triggering events are presented as events per 75 camera trap days; 75 days was the shortest deployment period (setup 3). A similar summary for birds (which were not the target of this study) can also be found in Barnett et al., 2022a (filename: 'REDFIRE_Trap_Camera_Summary_Birds_And_Other').


The mammalian species observed in the Red Forest (Table 1) included most of those observed in our other camera trapping studies across the CEZ (Wood & Beresford 2016). Exceptions were that we did not observe Brown bear (*Ursus arctos*), European bison (*Bison bonasus*) or Eurasian beaver (*Castor fiber*). The lack of these species in the Red Forest is to be expected:

•   The Red Forest did not contain suitable habitat for beaver during the study period.



- Photographic evidence of European bison in the Ukrainian CEZ was first recorded in 2015 at a site close to the Belarussian border (the species having been introduced into the Belarussian CEZ in 1996) (Gashchak et al., 2017); only one individual bull was recorded by camera traps 2015-2016 in the Ukrainian CEZ.

- The numbers of brown bear in the Ukrainian CEZ are low with no recorded sightings in the vicinity of the Red Forest at the time of this study (Gashchak et al., 2016).

A number of images recorded small groups of feral (domesticated) dogs which we have not observed elsewhere in the CEZ. It is likely that these are animals fed by workers at the nearby nuclear power complex. Images of so called 'stalkers' (illegal tourists) were also captured; these are not included in the dataset though they are identified in the image catalogue.


For mammals, Table 2 presents a summary by species and setup. For a number of species, Brown hare, Roe deer, Red deer, the number of triggering events was higher in setups 2 and 3; for Eurasian elk triggering were highest during setup 3 (Table 2). Whilst Wild boar and Przewalski's horse were observed during setups 1 and 2 none were recorded during setup 3. Observations of Eurasian lynx, European badger and Raccoon dog were lowest during setup 1. Young (new born) Eurasian

elk started to be observed in April/May 2017, with young Red and Roe deer being observed from June. Przewalski's horse were only observed in areas that had been burnt, potentially attracted by the new growth of grassy vegetation.





Table 2. Summary of medium/large mammal observations by setup.

| Species | Setup 1 Number of cameras species observed on | Setup 1 Mean number triggering events per 75d | Setup 1 Mean/ Maximum number of animals recorded per triggering event | Setup 2 Number of cameras species observed on | Setup 2 Mean number triggering events per 75d | Setup 2 Mean/ Maximum number of animals recorded per triggering event | Setup 3 Number of cameras species observed on | Setup 3 Mean number triggering events per 75d | Setup 3 Mean/ Maximum number of animals recorded per triggering event |
|---|---|---|---|---|---|---|---|---|---|
| Brown hare | 16 | 6.1 | 1.0/2 | 16 | 27.0 | 1.1/3 | 13 | 15.9 | 1.0/1 |
| Eurasian elk | 18 | 5.4 | 1.2/3 | 16 | 6.3 | 1.3/3 | 15 | 12.8 | 1.3/3 |
| Eurasian lynx | 6 | 0.6 | 1.2/2 | 2 | 1.5 | 1.0/1 | 1 | 2.0 | 1.0/1 |
| European badger | 3 | 0.5 | 1.0/1 | 4 | 2.7 | 1.0/1 | 2 | 1.5 | 1.0/1 |
| Feral dog (domesticated) | 4 | 1.0 | 2.1/4 | 6 | 1.5 | 6.0/4 | 3 | 2.0 | 1.7/5 |
| Grey wolf | 13 | 0.9 | 1.6/6 | 5 | 0.9 | 1.2/2 | 5 | 1.8 | 1.1/2 |
| Przewalski's horse | 6 | 1.5 | 1.2/5 | 5 | 1.5 | 1.2/3 | n/a | n/a | n/a |
| Raccoon dog | 2 | 0.8 | 1.0/1 | 8 | 2.3 | 1.0/2 | 8 | 3.4 | 1.0/2 |
| Red deer | 15 | 2.7 | 1.5/8 | 13 | 5.4 | 1.4/4 | 10 | 6.6 | 1.4/6 |
| Red fox | 9 | 2.1 | 1.1/2 | 9 | 3.9 | 1.0/1 | 7 | 3.1 | 1.0/1 |
| Roe deer | 15 | 1.4 | 1.3/4 | 16 | 8.1 | 1.1/1 | 13 | 7.1 | 1.2/3 |
| Wild boar | 5 | 0.8 | 2.0/6 | 6 | 1.4 | 2.5/7 | n/a | n/a | n/a |

Deployment periods were: Setup 1 September 2016 to March 2017; Setup 2 March 2017 – June 2017; Setup 3 June 2017 – September
2017. The total number of cameras operating in Setups 1, 2 and 3 were 18, 16 and 15 respectively. n/a not applicable, species not
observed.



## 4 Estimated weighted absorbed dose rates

Table 3 presents a summary of estimated total weighted absorbed radiation dose rates for the example large mammal and Red fox for each camera location assuming a home range of 0. 5 km radius and also over a radius of 5 km centred on the
middle of our study area. All mean, and most median, estimated dose rates are above the lower end of the International Commission on Radiological Protections (ICRP) Derived Consideration Reference Level (DCRL) for mammals of 1 mGy d⁻¹ (approximately 40 µGy h⁻¹) (ICRP 2008). The DCRLs are one order of magnitude dose rate bands, for mammals 1 – 10 mGy d⁻¹, within which radiation effects may be expected to occur.

Table 3. Estimated weighted absorbed dose rates to mammals comparing those estimated for a large mammal (a deer) and a relatively small mammal spending some time underground (Red fox). Estimates are presented for an area of 5 km radius centred on the Red Forest and also for an area of 0.5 km radius centred on each camera site.

| Site | Large mammal Total dose rate (µGy h⁻¹) Mean | Large mammal Total dose rate (µGy h⁻¹) SD | Large mammal Total dose rate (µGy h⁻¹) Median | Red fox Total dose rate (µGy h⁻¹) Mean | Red fox Total dose rate (µGy h⁻¹) SD | Red fox Total dose rate (µGy h⁻¹) Median |
|---|---|---|---|---|---|---|
| 5 km radius area | 47 | 123 | 17 | 40 | 100 | 16 |
| 155 | 94 | 188 | 44 | 81 | 150 | 40 |
| 156 | 197 | 366 | 99 | 168 | 281 | 89 |
| 157 | 90 | 166 | 45 | 78 | 131 | 41 |
| 158 | 132 | 270 | 61 | 113 | 216 | 55 |
| 159 | 413 | 675 | 227 | 348 | 512 | 205 |
| 160 | 171 | 363 | 77 | 145 | 282 | 70 |
| 161 | 293 | 543 | 146 | 247 | 412 | 132 |
| 162 | 448 | 725 | 247 | 377 | 550 | 223 |
| 164 | 95 | 222 | 40 | 80 | 171 | 37 |
| 165 | 386 | 620 | 215 | 324 | 469 | 193 |
| 166 | 262 | 463 | 136 | 221 | 349 | 123 |
| 167 | 183 | 299 | 100 | 154 | 226 | 90 |
| 170 | 182 | 322 | 97 | 150 | 233 | 86 |





| Site | Large mammal Total dose rate (µGy h⁻¹) Mean | Large mammal Total dose rate (µGy h⁻¹) SD | Large mammal Total dose rate (µGy h⁻¹) Median | Red fox Total dose rate (µGy h⁻¹) Mean | Red fox Total dose rate (µGy h⁻¹) SD | Red fox Total dose rate (µGy h⁻¹) Median |
|---|---|---|---|---|---|---|
| 171 | 131 | 211 | 72 | 109 | 158 | 65 |
| 172 | 105 | 177 | 57 | 86 | 129 | 50 |
| 173 | 164 | 282 | 87 | 134 | 204 | 78 |
| 174 | 55 | 132 | 23 | 46 | 98 | 21 |
| 175 | 40 | 98 | 29 | 33 | 71 | 15 |
| 364 | 277 | 452 | 152 | 230 | 336 | 137 |
| 365 | 130 | 270 | 60 | 109 | 199 | 55 |

In contrast to the 2009 snow track study of Møller & Mousseau (2013) we observed no reduction in the abundance of
mammals with increasing dose rate. The number of species observed at camera locations was relatively consistent with estimated median weighted absorbed radiation dose rate (Fig. 2). We also observed no relationship between estimated median weighted absorbed radiation dose and the number of triggering events for the main species observed (Brown hare, Eurasian elk, Red deer, Roe deer) (see examples from setup 3 in Fig. 3).




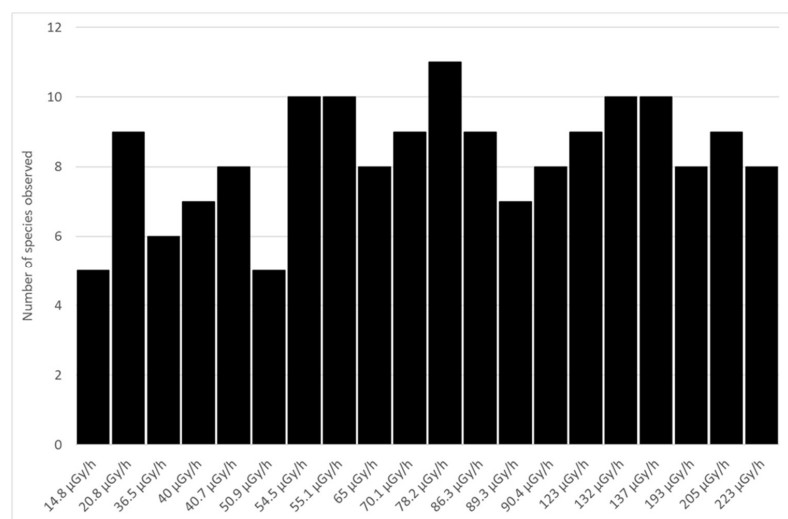

**Figure 2: Number of species observed by estimated weighted absorbed dose rate (note the estimated weighted absorbed dose rates presented are those estimated for the geometry approximating to a Red fox).**








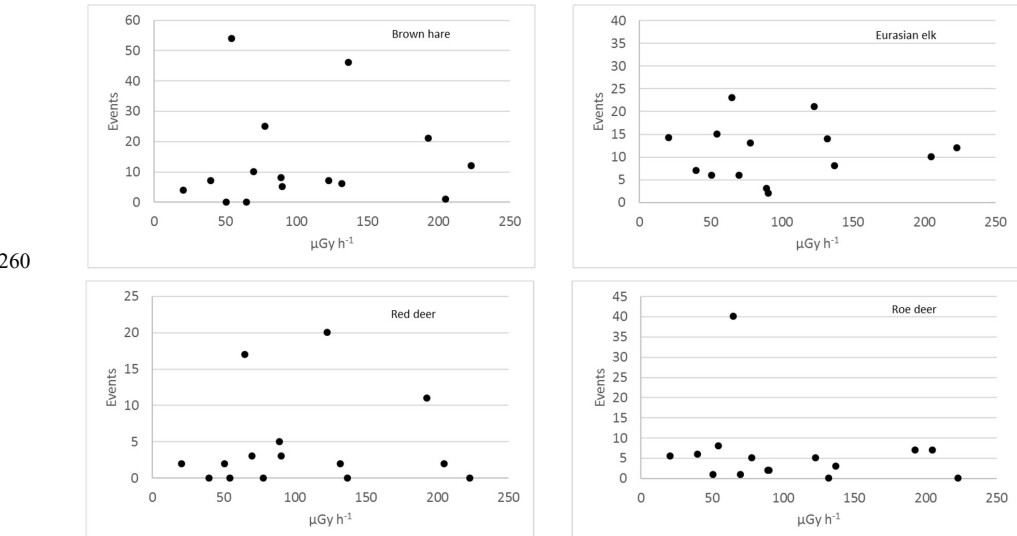

**Figure 3: Demonstration of the lack of relationship between number of triggering events and estimated absorbed weighted dose rates (using that calculated for the Red fox geometry as an example); data presented are from setup 3.**

**5 Data availability**

The data described here (https://doi.org/10.5285/bf82cec2-5f8a-407c-bf74-f8689ca35e83; Barnett et al. 2022a) are freely available from the Environmental Information Data Centre (https://eidc.ac.uk/, last access: 28 September 2022) under a Creative Commons Attribution (CC BY) licence.

**6 Applications of data**

The data presented will be of value to those studying wildlife within the CEZ both from the perspectives of the potential effects of radiation on wildlife and also rewilding in this large, abandoned area. Together with other trap camera data sets being published (e.g. Barnett et al., 2022b; Gashchak et al. 2022) the data will help in establishing a picture of wildlife across the CEZ. The data may also have value in any future studies investigating the impacts of recent Russian military action in the CEZ.



**Author contribution**

NAB and MDW secured funding for the study; NAB, MDW and SG defined the study protocols; SG conducted the fieldwork maintaining cameras and making field notes; CLB and SG, with input from NAB compiled and QC'd the image catalogue and accompanying documentation (Barnett et al., 2022a); NAB and CLB drafted the paper to which SG contributed and MDW reviewed.

**Competing interests**

The authors declare that they have no conflict of interest.

**Acknowledgements**

The authors are grateful to the following people for their assistance during the project: Eugene Guliaichenko (Chornobyl Center) for assistance during fieldwork and Claire Wells (UKCEH) for assistance with data entry. We also thank Jacky Chaplow (UKCEH) for proof reading the manuscript.

**Financial support**

This study was funded by the Natural Environment Research Council (NERC) as part of the RED FIRE project (Radioactive Environment Damaged by Fire: a Forest in Recovery, https://www.ceh.ac.uk/our-science/projects/red-fire-radioactive-environment-damaged-fire, last accessed 26 March 2022; grant no. NE/P015212/1).

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
