# Peer review of "Mammals in the Chornobyl Exclusion Zone's Red Forest: a motion activated camera trap study"

_Earth System Science Data, 2022_

## Author Response (AR1)

We thank both reviewers for the positive comments on our paper. Below we detail how we have responded to the specific points they raise:

RC 1

COMMENT>> My only suggestion refers to the Discussion of the results, and its comparison with a previous paper (Moller and Mousseau 2013). In my view, it would be worth to mention that the present study differs in many aspects to Moller and Mousseau (2013): different geographic scope (broader in the 2013 paper, which includes most of the Exclusion Zone, and not just the Red Forest), different contamination scenarios (2013 paper included areas with much lower radiation levels, which may affect the comparisons regarding the effects of radio-contamination on mammal distribution)…

RESPONSE>> We have addressed this comment directly the first time the Moller and Mousseau paper is mentioned in the introduction. We have also added reference to three studies which show, over the wider CEZ, no relationship between mammal abundance/diversity and radiation exposure. In the Discussion, we now state our data do not support the low number of mammals in the Red Forest as reported by Moller and Mousseau.

COMMENT>> I would prefer to have Table 1 arranged following a phylogenetic order, rather than an alphabetic one.

RESPONSE>> We think for most readers presenting alphabetically is an easier way to consult the table.

COMMENT>> Figure 3. I suggest to remove "Demonstration of the lack of" from the legend.

RESPONSE>>Edited as suggested

RC2

COMMENT>> One suggestion is to add additional details on the dose calculations. External dose is not addressed explicitily. I assume external dose is simulated using the ERICA model and based on radioactivity soil concentrations of 137Cs and 90Sr. If so, please add what soil moisture variable was used in your ERICA model run. External dose simulated by ERICA is quite sensitive to soil moisture, and this information could be useful to others that want to use your data. You do state that ambient external dose rate was measured at each camera trap location. Please make it clear to the reader whether the ambient dose rate was used to estimate dose rate to wildlife, or if the ERICA model simulated both internal and external dose rates.

RESPONSE>>The ERICA Tool was used to estimate external dose rates and this is now clearly stated in the manuscript. We assumed 100% soil dry matter and this is now noted in the section on dose estimation; we also site soil dry matter contents from other studies in the Red Forest to put this value into context.